# Recent Progresses in Pyrolysis of Plastic Packaging Wastes and Biomass Materials for Conversion of High-Value Carbons: A Review

**DOI:** 10.3390/polym16081066

**Published:** 2024-04-11

**Authors:** Youliang Cheng, Jinpeng Wang, Changqing Fang, Yanli Du, Jian Su, Jing Chen, Yingshuan Zhang

**Affiliations:** 1Faculty of Printing, Packaging Engineering and Digital Media Technology, Xi’an University of Technology, Xi’an 710048, China; chengyouliang@xaut.edu.cn (Y.C.); 18729187396@163.com (J.W.); sujian@xaut.edu.cn (J.S.); chenjing@xaut.edu.cn (J.C.); 2220821145@stu.xaut.edu.cn (Y.Z.); 2Shaanxi Zhonghe Dadi Industrial Limited Company, Xianyang 712099, China; dyl19860804@126.com

**Keywords:** plastic packaging wastes, co-pyrolysis, polyolefin, carbon nanotubes

## Abstract

The recycling of plastic packaging wastes helps to alleviate the problems of white pollution and resource shortage. It is very necessary to develop high-value conversion technologies for plastic packaging wastes. To our knowledge, carbon materials with excellent properties have been widely used in energy storage, adsorption, water treatment, aerospace and functional packaging, and so on. Waste plastic packaging and biomass materials are excellent precursor materials of carbon materials due to their rich sources and high carbon content. Thus, the conversion from waste plastic packaging and biomass materials to carbon materials attracts much attention. However, closely related reviews are lacking up to now. In this work, the pyrolysis routes of the pyrolysis of plastic packaging wastes and biomass materials for conversion to high-value carbons and the influence factors were analyzed. Additionally, the applications of these obtained carbons were summarized. Furthermore, the limitations of the current pyrolysis technology are put forward and the research prospects are forecasted. Therefore, this review can provide a useful reference and guide for the research on the pyrolysis of plastic packaging wastes and biomass materials and the conversion to high-value carbon.

## 1. Introduction

Currently, there is growing global concern for the balanced development of the economy and the environment. Plastic products constitute a significant portion of the packaging industry [1]. Due to its durability, light weight, and low cost, it is widely used in the storage and packaging of consumer goods, such as electronics, medicine, chemicals, food, and beverages [2,3]. For instance, the demand for plastic packaging products has increased in recent years in China, resulting in the production of approximately 18 million tons of plastic packaging wastes and causing serious environmental pollution. If a large amount of plastic packaging waste enters the ecosystem, it can have extensive detrimental effects on the air, soil, and water environment. This can lead to the formation of microplastics, which can eventually pose a direct hazard to human health [4]. If these plastic wastes can be effectively recycled, it not only reduces the pressure on the environment caused by pollution, but also obtains high-value resources such as hydrogen, methane, light oil, and carbon materials to save fossil fuels [5].

Plastic packaging waste is primarily composed of polyolefins and polyethylene terephthalate (PET). Polyolefins include high-density polyethylene (HDPE), low-density polyethylene (LDPE), polypropylene (PP), polystyrene (PS), and polyvinyl chloride (PVC) The largest proportion of plastic packaging waste comes from waste polyethylene (WPE), waste polypropylene (WPP), and waste polystyrene (WPS) [6,7]. Disposing of plastic wastes can be achieved through various methods, such as landfill, incineration, recycling, photodegradation, and chemical recycling. While traditional landfill and incineration methods are simple and efficient, they can also cause secondary pollution to the land and atmosphere. The mechanical recycling process involves decontamination, remelting, remolding, re-extruding, and recasting. This process is limited to targeting a single type of pollution-free plastic waste. However, the mechanical stability of recycled plastics is typically reduced after each cycle [8,9]. The use of chemical additives such as fillers, plasticizers, colorants, stabilizers, and forming agents is often necessary in plastic packaging applications. This results in a complex composition, making mechanical and chemical recycling of plastic waste containing flexible plastic packaging technically challenging on a large scale [10].

Various methods of chemical recycling exist, including direct pyrolysis, catalytic pyrolysis, hydrolysis, and alcoholysis [11]. Pyrolysis is a thermochemical conversion technology that can convert plastic packaging wastes into high-value products in an environmentally sustainable manner. This method has received significant attention [12]. Due to the excellent properties of carbon materials, such as high surface area, porosity, electronic conductivity, rich/tailorable surface chemistry, and structural stability at high temperatures, converting plastic packaging wastes into high-value carbon materials via pyrolysis is a promising strategy [13,14,15]. As a result, these carbon materials can be utilized in various fields such as adsorption, conductivity, supercapacitors, lithium-ion batteries, biomedicine, and others [16,17,18,19]. In addition, carbon materials are the second most commonly used type of nanomaterials, surpassed only by metal nanoparticles [20].

Biomass is a valuable natural resource with high utilization potential. The agricultural sector produces approximately 140 billion tons of waste biomass annually, but only 40 wt% of it is reused [21]. Every year, forestry and agricultural residues produce approximately 181.5 billion tons of lignocellulosic biomass, but only 4.5 wt% of this waste is recycled [22]. Instead of disposing of or burning a significant amount of biomass, it is valuable to convert this resource into high-value carbon to maximize the potential of renewable energy [23]. Numerous studies have demonstrated that biomass materials, typically containing up to 55% carbon, represent a significant source of renewable energy and carbon [24]. The carbon content increases as other elements are removed from the biomass through thermochemical conversion. Biochar is a high-value product that has been used to reduce greenhouse gas emissions, improve soil nutrients, and prepare energy storage devices [25,26,27]. Singh et al. [28] utilized biochar and plastic char derived from biomass materials and waste plastics, respectively, to eliminate heavy metals from aqueous solutions. The specific surface area of biochar was significantly higher than that of plastic char and increased with increasing pyrolysis temperature, confirming that biochar is a more suitable adsorbent [29,30].

Because biomass has low hydrogen and carbon contents, resulting in low carbon efficiencies and high coke formation during pyrolysis, an alternative approach to overcome these limitations is to co-pyrolyze biomass with plastics, which are rich in hydrogen and carbon contents [31]. Several studies have shown that co-pyrolysis requires less activation energy than pyrolysis of a single feedstock. Co-pyrolysis has been found to enhance the quality of both carbon materials and bio-oil when compared to the pyrolysis of separate components [32]. Currently, the co-pyrolysis of plastic wastes and biomass materials is primarily focused on producing liquid products, such as bio-oil, and gas products, such as olefins, aromatics, and hydrogen. However, the resulting pyrolysis oil and gas face several issues, including poor quality and high costs associated with separation and purification. These issues limit the widespread commercial application of these pyrolysis products [33]. Stable and high-value carbon materials can be produced through the co-pyrolysis of waste plastics and biomass materials, making it a promising strategy.

Therefore, optimizing the high-value conversion route of packaging waste plastics and biomass materials is crucial. This review summarizes the routes of pyrolysis for typical polyolefin materials, PET, and typical biomass materials to produce high-value carbons and related applications. By comparing the relevant literature, this review summarizes recent progress in the field, identifies limitations of pyrolysis technology, and proposes research prospects for carbon materials derived from the pyrolysis of packaging waste plastics.

## 2. Co-Pyrolysis of Plastic Wastes and Biomass Materials

Co-pyrolysis is a process in which multiple feedstocks are thermally decomposed in the absence of air or oxygen. Co-pyrolysis has many advantages over individual pyrolysis due to the synergistic effects between different feedstocks, which can improve the quality of pyrolysis products and reduce costs due to the ability to process mixed plastic waste and biomass materials [34,35].

The synergistic effect is very important in the co-pyrolysis process. Due to the interaction of multiple raw materials and different process parameters during co-pyrolysis, the synergistic mechanism is very complicated. Positive or negative synergy is influenced by pyrolysis time, temperature, heating rate, and catalyst properties, but the most important factor is the type of feedstock [36,37]. At present, the research on the co-pyrolysis of plastic wastes and biomass can improve the yield and quality of liquid products such as bio-oil, which has been systematically reviewed. Usually, carbon materials are considered as by-products during the co-pyrolysis because of their low yield [38]. Therefore, it is a promising area of research to improve the yield and quality of carbon products by utilizing the synergistic mechanism during the co-pyrolysis of plastic waste and biomass materials. The proposed co-pyrolysis mechanism of biomass and plastics is shown in Figure 1 [39].

In this section, we will summarize the different routes by which high-value carbons have been produced from the co-pyrolysis of plastic waste and biomass materials. These high-value carbons mainly include biochar, graphene, graphitic carbon, activated carbon, and carbon nanotubes.

### 2.1. Biochar

Biochar is a common product during the co-pyrolysis of waste plastics and biomass materials, which is rich in inert carbon and nutrient elements and has a large specific surface area and rich functional groups [40]. The properties mentioned are crucial for maximizing the value of the material and achieving a synergistic effect through co-pyrolysis, which can enhance the yield and carbon stability of biochar [41,42]. Compared to commercial charcoal, which has a heating value of approximately 30 MJ/kg, biochar has a lower heating value ranging between 13.4 and 17.40 MJ/kg [43,44]. The low heating value of biochar is mainly due to the low heating value properties of the raw biomass itself. Therefore, the quality of biochar can be improved by co-pyrolyzing biomass with high calorific value materials such as plastic (41.7–46.4 MJ/kg) [45].

The yield of biochar from co-pyrolysis of plastic wastes and biomass materials is primarily determined by the co-pyrolysis temperature and the blending ratio of the feedstocks. Rathnayake et al. [46] conducted the co-pyrolysis of waste agricultural plastic (LDPE) and biomass, mixing them with plastic ratios of 0, 0.25 wt%, 2.5 wt%, 5 wt%, and 10 wt%. They found that there was a slight increase in yield for the system with 0.25 wt% and 2.5 wt% (5.3 wt% and 3.4 wt% increase, respectively) compared to 0. However, biochar yield decreased by 1.6 wt% and 4.8 wt% in 5 wt% and 10 wt%, respectively. The average biochar yield was 50 ± 2 wt%. Berthold et al. [47] studied the co-pyrolysis of WPP, WPS, waste PET, polycarbonate (PC), and rice husk (RH) at 600~800 °C and then analyzed the influencing factors of biochar yield. They found that plastics yielded less biochar than RH at the same temperature, and the yield of biochar was between 20 wt% and 30 wt%. Wantaneeyakul et al. [45] investigated that the co-pyrolysis process of waste HDPE and RH at 400, 500, and 600 °C. When the HDPE mass fraction increased from 10 wt% to 50 wt%, the biochar yields decreased from 36.53 wt% to 22.91 wt% at 500 °C and from 36.95 wt% to 22.42 wt% at 600 °C. Increasing the HDPE mass fraction initially resulted in a decrease in biochar yield at a pyrolysis temperature of 400 °C. Specifically, the yield decreased from 41.36 wt% to 33.02 wt% when the HDPE mass fraction increased from 10 wt% to 30 wt%.

Several studies have reported the influence of pyrolysis temperature on biochar yield [48,49]. In general, biomass materials require a higher pyrolysis temperature than plastic wastes. When the pyrolysis temperature is lower than the complete degradation temperature of the plastic, volatile substances from the pyrolysis of biomass materials become trapped in the molten plastic, resulting in coking [50]. However, the yield of biochar decreased as the pyrolysis temperature increased because plastic wastes and biomass materials are mostly degraded into volatile substances or small gas molecules at high pyrolysis temperatures. The negative synergistic effect on biochar production may be due to the interaction between biomass and polymer, which tends to produce more volatile products. Initially, the free radicals from the decomposition of biomass materials promote the degradation of plastic wastes and the formation of hydrocarbon radicals [51]. This interaction involves the initiation of secondary radical formation through depolymerization, hydrogen transfer, and the reaction between radicals [52].

In recent years, there have been many reports regarding the use of biochar as an adsorbent [53]. Biochar, which typically contains oxygen-containing functional groups, has been considered a promising low-cost adsorbent compared to activated carbon [54,55]. It facilitates the adsorption of pollutants through hydrogen bonding, electrostatic mechanisms, and surface complexation [56,57]. Xu et al. [58] prepared brominated biochar by the co-pyrolysis of brominated flame retarded (BFR) plastic and waste biomass, which can remove elemental mercury from coal-fired flue gas. This strategy could combine waste disposal and mercury sorbent preparation in one process and the mercury adsorption capacity of prepared biochar was close to that of commercial activated carbons. Bernardo et al. [59] found that biochar obtained after the co-pyrolysis of pine, used tires, and plastic wastes at 420 °C had more mesopores and oxygen functional groups and exhibited a high adsorption capacity for Pb^2+^. Silori et al. [60] used biochar by the co-pyrolysis of sawdust and plastic wastes to adsorb antibiotics in wastewater, and the adsorption capacity of biochar obtained by co-pyrolysis was lower than that obtained by individual pyrolysis of biomass.

Meanwhile, many studies have reported the use of biochar obtained from co-pyrolysis of biomass and plastic waste in agriculture and fuel. It is a promising approach due to its potential to reduce waste and improve soil quality [61]. Yao et al. [62] investigated the co-pyrolysis of biomass, low-quality coal, and waste agricultural film at 550 °C and the yield of biochar was 55.6 wt%. Han et al. [63] found that biochar obtained by the co-pyrolysis of straw and WPE at a ratio of 5:1 promoted the development of plant roots and increased soil water and oxygen content effectively. Peng et al. [64] found that the porous structure and active functional groups appeared in the biochar after the low-temperature co-pyrolysis of food waste and waste PVC. Xu et al. [65] used the metal-doping mesoporous graphite-like catalysts as the catalyst during the co-pyrolysis of corn stover and plastic wastes, and the biochar with more agglomerative, and smoother morphologies exhibited some shallow pores. Luo et al. [66] studied the co-pyrolysis of waste plastics (PE, PP, PS, PVC) and biomass materials (food waste and phycocyanin) at 700 °C and analyzed the inhibition effect on cyanide. They found that the co-pyrolysis of waste PE, PP, and PS with biomass materials had few effects on the inhibition of cyanide, while the concentration of cyanide in carbon products decreased after the co-pyrolysis of biomass materials with PVC.

### 2.2. Graphene

Graphene is composed of a honeycomb sheet of carbon atoms with a sp2 hybridized structure arranged on the same plane at 120 °C bond angles [67]. The addition of graphene to composite materials significantly enhances their thermal, mechanical, and electrical properties, making it a promising material [68,69]. In the past, graphene was prepared through physical exfoliation, chemical exfoliation, or reduction of graphene oxide. However, these methods are expensive and limit the production of high-quality graphene or graphene nanoplatelets on a large scale [70].

The production of graphite requires extremely high temperatures, making the traditional low-temperature co-pyrolysis technology unsuitable. Recently, a new type of pyrolysis technology called flash Joule heating (FJH) has been reported. FJH utilizes instantaneous high-current heating to raise the temperature of gram-grade raw materials up to 4000 °C within one second [71]. Several researchers have demonstrated that the FJH process can convert carbon-containing feedstocks, such as plastic wastes, rubber, and biomass materials, into high-quality graphene. This is conducive to the large-scale commercial application of graphene. The carbon product obtained through FJH can be easily exfoliated, making it a valuable material. During the process of FJH for plastic wastes and biomass materials, a 5 wt% carbon black is added to the mixed feedstocks to improve conductivity. The mixture is then placed in a flashing chamber, such as a tubular or flat reactor (as shown in Figure 2). Finally, graphene is produced through graphitization by exposure to high-power FJH, promoted by a direct current (DC) pulse discharge without the use of a catalyst. During the rapid procedure, non-carbon atoms in the mixture sublimate as small molecules, resulting in the production of carbon materials with high carbon content. This process also eliminates the negative impact of the complex composition of mixed plastic wastes on carbon products [14].

Recently, most research on the FJH is still focused on the individual pyrolysis of plastic wastes. Luong et al. [71] proposed that the FJH of inexpensive carbon sources such as biochar, biomass, carbon black, and mixed plastic wastes could obtain gram-scale quantities of graphene in less than one second. The carbon product after the FJH was named flash graphene (FG), and it showed turbostratic arrangement (that is, little order) between the stacked graphene layers. The disordered orientation of FG layers facilitated its rapid exfoliation upon mixing during composite formation, and Raman spectroscopy analysis shows a low-intensity or absent D band for FG, indicating that FG has among the lowest defect concentrations reported so far for graphene. Algozeeb et al. [72] studied the FJH of plastic wastes (HDPE, LDPE, PP, PS, PVC, and PET) through alternating current (AC) and direct current (DC) pulse discharge, and about 22 wt% yield of high-quality flash graphene could be obtained. When the particle size of plastic wastes was in the range of 1–2 mm, the yield of flash graphene increased as thermal stability increased and the resistivity decreased. Wyss et al. [73] investigated the pyrolysis ash from the FJH of WPP, and they found that the FG yield generally achieved was 85~90 wt% after FJH, which could be easily exfoliated. In addition, FG was added to polyvinyl alcohol (PVA) to prepared nanocomposites. Compared to neat PVA, the addition of FG improved the mechanical properties and barrier properties of the nanocomposites. Figure 3a–f show representative images of the wrinkled graphene and large tFG crystals, respectively. Advincula et al. [74] studied the FJH of waste HDPE, and the process yield of FG was about 40 wt%. FG as an ideal candidate for lubricating oil additives effectively reduced friction and wear. Additionally, the FG production by FJH reduced water and energy consumption compared to conventional production of graphene. Wyss et al. [75] used calcium acetate (Ca (OAc)_2_) as the activator during the FJH of plastic packaging wastes including carbonated beverage bottles, milk jugs, grocery bags, food packaging, and coffee cups, and the holey and wrinkled FG with high surface area and defect concentration was found to be effective as a metal-free hydrogen evolution reaction electrocatalyst, Li-metal battery anode, and CO_2_ gas adsorption material. Advincula et al. [76] found that FG obtained after the FJH of WPE and amorphous carbon (AmoC) derived from CO_2_ had a maximum yield of 50 wt%, which could enhance the mechanical properties of composite materials as an additive.

### 2.3. Graphitic Carbon

Graphitic carbon is composed of loosely bound graphene layers stacked together into a hexagonal structure. It is considered one of the most stable and strongest materials, capable of withstanding very high temperatures without damaging its structure. Graphitic carbon can be produced from biochar derived from organic wastes, including animal, agricultural, or plastic wastes [67]. In comparison to biochar, graphitic carbon exhibits greater electrical and thermal conductivities [67].

#### 2.3.1. Non-Catalytic Co-Pyrolysis

During the co-pyrolysis of waste plastics and biomass materials, the percentage of every kind material can greatly affect the yield and properties of carbon materials. Lu et al. [77] conducted the co-pyrolysis of WPE, waste PVC, and pine wood (PW), mixing PW-PE and PW-PVC with mass ratios of 1:1, 1:3, and 3:1. They found that the oil yield increased and the coke yield decreased by 13.8~22.4 wt% after the co-pyrolysis of PW-PE compared to the calculated results. For the co-pyrolysis of PW-PVC, the coke yield increased by 15.5~27.9 wt% compared with the calculated results, while the oil yield decreased. Compared with individual pyrolysis, the specific surface area of coke by the co-pyrolysis of PW-PE increased, and that by the co-pyrolysis of PW-PVC reduced. Xue et al. [78] found that carbon materials obtained after the co-pyrolysis of waste HDPE and red oak at 625 °C had the maximum calorific value. In addition, the specific surface area of the carbon products decreased when the temperature increased and many large pores were obtained through the synergy between red oak and molten HDPE at a high temperature. Bernardo et al. [79] found that adding waste tires to the co-pyrolysis system of pine and waste plastics (56% (*w*/*w*) PE, 27% (*w*/*w*) PP, and 17% (*w*/*w*) PS) can increase the coke yield. A sequential extraction with solvents of increasing polarity provided a removal of the pyrolysis liquid-phase products from the co-pyrolysis char, and the total extraction yields of the sequential extractions were 81 wt% for char. The upgraded chars are mainly mesoporous and macroporous with significant adsorption capacity for the bulky molecule methylene blue.

In addition, PET occupies a high proportion in drinking water packaging and transparent packaging boxes. However, the recovery rate of waste PET is very low, resulting in a large amount of solid waste [80,81]. Ko et al. [82] revealed that waste PET was almost completely converted into carbon materials during the co-pyrolysis of macadamia nut shell and coconut shell with PET at 450 °C. In the case of blends, the pyrolyzed degree of PET increased with the increase in pyrolysis temperature and biomass material content in blends. Furthermore, the porosity of chars can vary with the rate of thermal degradation depending on the raw materials, and high hemicellulose and cellulose fractions lead to highly constricted chars with a high portion of microporosity.

#### 2.3.2. Catalytic Co-Pyrolysis

To the best of our knowledge, the preparation of graphite typically requires an extremely high temperature of 3000 °C. The use of catalysts can significantly reduce the graphitization temperature of various raw materials to below 1000 °C. This process can also enhance the quality of graphitic carbon and increase the carbon content of the resulting products. Common catalysts for reducing the formation temperature of graphitic carbon include Mg, Fe, Co, Ti, Mn, and Ni metals [83]. Transition metals, such as iron, titanium, and manganese, can also improve the porosity, specific surface area, and electrical conductivity of graphitic carbon.

Ryu et al. [84] used MgO impregnated with C, Al_2_O_3_, and ZrO_2_ as the catalyst during the co-pyrolysis of cellulose and waste LDPE, and they found that the graphitic carbon yield was higher than that using co-pyrolysis with MgO/Al_2_O_3_ and MgO/ZrO_2_ as the catalyst. Chattopadhyay et al. [85] revealed that fold structure appeared in the products of graphene oxide (GO) when cobalt-based catalysts were used for the catalytic co-pyrolysis of biomass materials and waste plastics (HDPE, PP, and PET). In addition, the yield of carbon products decreased with the increase in pyrolysis temperature and the catalyst with the highest content of cobalt had the lowest yield of carbon products. Luo et al. [86] studied the co-pyrolysis of waste masks composed of PE, PP, nylon, and bio-oil at different temperatures (700, 800, and 900 °C), and then the obtained graphitic carbons were catalyzed with Ni/GF composites. They found that the biomass carbons had the highest carbon content and can be used as a potential substitute for anthracite, where its ash content was between walnut shell and anthracite. Figure 4a–d show that the high-quality three-dimensional graphene films (3DGFs) with porous structure obtained at different temperatures had a 3D network without significant cracks or collapse, and indicate that 3DGFs consist of numerous interconnected pores, which provide a large surface area and volume for its practical application in the fields of oil spill cleanup and oil/water separation.

Kim et al. [87] investigated the catalytic co-pyrolysis of boxwood and HDPE using HZSM-5 and Al-MCM-41 as the catalyst, respectively. They found that the Al-MCM-41 catalyst promoted the formation of more chars (29 wt%) compared with HZSM-5. Zheng et al. [88] used HZSM-5 as a catalyst during the co-pyrolysis of Yunnan pine and LDPE at 500 °C, and the carbon product yield was higher than non-catalytic pyrolysis and individual pyrolysis of biomass and plastics. Zhang et al. [89] found that carbon materials obtained after the catalytical co-pyrolysis of biomass materials (cellulose/Douglas fir sawdust) and waste LDPE by ZSM-5 significantly reduced the pyrolysis temperature and provided an energy-saving route for the co-pyrolysis. Johansson et al. [31] studied the co-pyrolysis of different waste plastics (PET, PP, PE, and PVC) with wooden biomass using HZSM-5 as the catalyst, which not only reduced the reaction temperature but also promoted the formation of coke.

Sanjana et al. [90] prepared carbon products with high calorific value by co-pyrolysis of cellulose and LDPE, and the recovery rate of total energy reached 92%. Sajdak et al. [91] revealed that the higher carbon content and dehydrogenation degree appeared in the carbon products by the co-pyrolysis of WPP, alder, and pine. The high ratios of C/H and C/O proved that this process had a good prospect of synthesizing high-quality graphitic carbon.

### 2.4. Activated Carbon

In recent years, solid adsorbents such as activated carbon (AC) have attracted considerable attention. ACs have been widely studied as potential alternative materials due to low regeneration energy, good thermal and mechanical stability, large porosity, adjustable porosity, low cost, and easy regulation of surface chemical properties [92,93]. Compared to biochar, activated carbon has a higher adsorption capacity. A high weight loss rate during the co-pyrolysis of biomass and plastic wastes is beneficial for preparing ACs and improving the adsorption capacity of the products. Additionally, the surface properties of ACs were enhanced through treatment with activators, resulting in increased adsorption capacity [94].

Li et al. [95] used K_2_CO_3_ as the activator during the co-pyrolysis of cyanobacteria and WPP, and the physical, chemical, and adsorption properties of ACs were affected by the pyrolysis temperature, activator’s ratio and PP content. After the activation, the pore volume and surface area of ACs increased (up to 2811 m^2^/g and 1.992 cm^3^/g, respectively) and exhibited a high adsorption capacity for organic pollutants. Compared with PP-free activated carbon, the surface area of ACs increased 3.1 times. Gale et al. [96] revealed that the high adsorption capacity appeared for the carbon materials with low surface area but large average pore size when KOH was used for the activation during the co-pyrolysis of corn stover (CS), WPS, and waste PET, while the evolution of gas byproducts from the cracking of polymers affected ACs’ crystallographic structure, pore structure, pore size, and surface area (as shown in Figure 5). The ACs from CS:PET 1:1 had the highest measured surface area at a 1:1 ratio, 423.8 ± 24.8 m^2^/g. Gopu et al. [97] studied the co-pyrolysis of municipal solid waste including waste plastics, paper, wood, and food using CO_2_ as the activator. They found that the mass yield of ACs decreased from 30 wt% to 11 wt% when the temperature increased from 408 °C to 900 °C. However, removal of volatile substances promoted the increase in surface area and porosity for ACs at a high temperature. The obtained ACs showed surface areas over 300 m^2^/g and had a high methylene blue adsorption capacity and could be applied in water treatment.

Compared with biochar through hydrogen bonding and electrostatic adsorption, ACs mainly rely on pore structure and physical adsorption [98,99], which can effectively remove color, odor, organic matter, and other pollutants in wastewater. Li et al. [100] found that ACs from the co-pyrolysis of the apple tree pruning waste and PET with mass ratio of 1:1 had a carbon yield of about 26 wt%. The specific surface area of the activated carbon was 1808 m^2^/g, the total pore volume was 0.737 cm^3^/g, and the micropore volume was 0.674 cm^3^/g, which exhibited high selectivity to CO_2_ and excellent stability even after multiple adsorption cycles. Martín-Lara et al. [101] found that ACs obtained after the co-pyrolysis of WPE, WPP, and WPS at 550 °C had the highest lead adsorption capacity, and the yield of solid product was the highest at 450 °C, reaching almost 14 wt%. As it happened with the specific surface area, the pore volume with the highest value, being 166.76 Å for 450 °C (approximately 3 times smaller for 550 °C), increased with increasing pyrolysis temperature, which demonstrated that a higher pyrolysis temperature improved the properties of the solid product as an adsorbent. Nistratov et al. [102] investigated the co-pyrolysis of WPP, waste LDPE, waste PET, and wood, and they found that the optimal pyrolysis condition was a pyrolysis temperature of 370 °C at a heating rate of 2.5 °C/min. The yield of ACs was 53 wt% and they had good adsorption effect on benzene vapors. These studies not only demonstrated a promising approach to address the environmental issue of polymeric plastic wastes but also created highly effective adsorbents.

### 2.5. Carbon Nanotubes

During the co-pyrolysis process of biomass and plastic wastes, a significant proportion of the resulting pyrolysis gas is composed of permanent gas and C2–C6 hydrocarbons. This gas has a high heating value and complex components and is typically combusted to provide heat [103]. However, this approach not only generates substantial CO_2_, exacerbating the greenhouse effect, but also overlooks the further value of the syngas. Therefore, it is a promising research area to obtain high-value products derived from syngas from an economic point of view [104,105].

Carbon nanotubes (CNTs) have excellent electrical conductivity, mechanical strength, and thermal conductivity, and have been used in adsorption materials, semiconductor transistors, energy storage, and so on [106,107,108]. Chemical vapor deposition (CVD) is one of the main methods for the synthesis of CNTs, where hydrocarbon gases (e.g., alkanes, alkenes, and alkynes) or syngas consisting of the mixture of CO and H_2_ are used as precursors for industrialized production of CNTs [109,110]. Moreover, the co-pyrolysis of biomass and plastics not only improves the quality of pyrolysis gas, but also promotes the growth of CNTs. Providing precursor materials for CNTs through the co-pyrolysis of waste plastics and biomass has been of interest to researchers [111]. Because the yield and quality of CNTs are influenced by co-pyrolysis or catalytic conditions, the two-stage pyrolysis method is often used to produce high-quality CNTs, where the most important point is to develop efficient supported metal catalysts [112]. The yield morphology and quality of CNTs depend on the active metal in the catalysts. Transition metals Fe, Co, Ni, and their compounds have been proved to be the best catalysts for the synthesis of CNTs [113]. The transition metal loaded on a porous support as the catalyst can take advantage of the catalytic activity of the metal and the high specific surface area and shape selection of porous support. Furthermore, the literature has shown that the supported bimetallic catalysts possess higher catalytic activity, promoting the thermal cracking of feedstock and the formation of light aromatic hydrocarbons and hydrogen gas [114,115].

The two-stage catalytic pyrolysis reactor is a common piece of equipment for co-pyrolysis of CNTs and contains four parts, including gas provider, heating system, liquid condenser, and gas collection system. The pyrolysis volatiles produced in the vertical reactor from biomass materials and plastic wastes pass to the liquid condenser for collecting pyrolysis oil, which ensures that only non-condensable gases enter the horizontal reactor for catalytic synthesis of carbon materials.

Dong et al. [116] found that multi-walled carbon nanotubes (MWCNTs) obtained after the co-pyrolysis of oil tea camellia shell and WPP in a two-stage reactor at 600~750 °C had high yield and some complete or incomplete bamboo-like knots by using NiCoCe@Al catalyst. The MWCNTs reveal some hollow tips, indicating tip growth, and this mode is more conducive to the growth of CNTs [117]. Xu et al. [118] investigated the catalytic co-pyrolysis of WPE and rice husk by using Ni/Al_2_O_3_ as the catalyst in a bench-scale fixed-bed reactor at 600 and 800 °C, respectively, and they found that straight and long CNTs with high purity and quality were obtained, along with uniform diameters (~16 nm). In addition, the yield of CNTs with hollow tubular structure was 139 mg/g-cata. Figure 6 shows SEM and TEM morphologies of the carbon products on Ni/Al_2_O_3_ catalysts in different PE proportions after catalytic upgrading of pyrolysis gas. Dong et al. [119] studied the influences of six catalysts Ni@MgO, Fe@MgO, Co@MgO, NiMo@MgO, FeMo@MgO, and CoMo@MgO (Ni, Fe, Co, and Mo as active metal, MgO as catalyst support) on the formation of CNTs by the co-pyrolysis of WPP and oil tea camellia shells in a two-stage reactor at 600~800 °C. Compared with monometallic catalyst, the bimetallic catalysts with Mo showed better catalytic performance. For example, the growth of MWCNTs with ultrafine tube diameters via CO disproportionation reaction on NiMo@MgO, FeMo@MgO, and CoMo@MgO exhibited excellent graphitization and the functional carbon nanocomposites obtained by adding the prepared MWCNTs exhibited good sterilization performance. Interestingly, Luo et al. [120] found that more high-quality CNTs with a smoother tube wall were obtained after the co-pyrolysis of WPP and herb residue in the two separate quartz tube reactors at 600~800 °C and catalyzation by metal/biochar catalyst. The strong secondary cracking capacity of NiCu-C promoted the formation of CH_4_ and CO, which provided an adequate carbon source for the generation of more high-quality CNTs at the back end.

At present, there are few studies on the co-pyrolysis of biomass and plastic wastes to obtain CNTs, mainly focusing on the co-pyrolysis of mixed waste plastics. Panahi et al. [121] investigated the catalytic pyrolysis of WPE, WPP, WPS, and waste PET in a two-stage quartz tube at 800 °C by using a stainless-steel substrate as the catalyst, and they found that pickling and air heat treatment of the catalyst promoted the yield of CNTs with multiwall characteristics. In addition, the yield of CNTs followed an order according to the specific waste plastic: PP > PE > PS > PET, and gaseous hydrocarbons produced by the pyrolysis of WPP and WPE proved to be effective carbon resources for CNTs. The CNT lengths (in the range of 3–20 μm), diameters (in the range of 20–100 nm), and wall thickness were found to depend on the polymer feedstock, on the type of wire-cloth catalyst substrate, and on the pretreatment of the catalyst. Wang et al. [122] studied the influence of catalyst, including iron, nickel, and magnesium, and reaction pressure on the formation of MWCNTs by the pyrolysis of WPP at 500~750 °C. The yield of MWCNTs with outer diameters between 10 and 30 nm and inner diameters around 5 nm was the highest when using nickel catalyst, and the filamentous carbons occupied about 93 wt% in the products. Furthermore, high reaction pressure was beneficial to the yield of CNTs; however, excessive pressure inhibited the growth of CNTs and resulted in the formation of short CNTs with a large diameter.

Veksha et al. [123] revealed that MWCNTs appeared in the carbon products with tangled shape when the catalyst was prepared by loading NiO on CaCO_3_ for the catalytic co-pyrolysis of flexible plastic packaging waste at 400~700 °C, which was composed of flexible monolayer films such as PE, PP, and PA. However, MWCNTs from flexible plastic packaging potentially replace conventional electrode materials. Jiang et al. [124] studied the co-pyrolysis of the disposable face mask in a dual-stage fixed-bed system at 400~800 °C, which was made of non-degradable synthetic polymers such as PP, PE, and PA, and CNTs with the diameter of ~40 nm via dense entangled growth were obtained by catalyzing with the metal-doped carbonaceous catalysts (M/Cs). Zhu et al. [125] used Fe@Al_2_O_3_ as the catalyst during the co-pyrolysis of waste plastics (WPE, WPP, WPS, and waste PET) in a two-stage reactor at 500~800 °C, and the carbon yield and CNT purity can reach 32.21 wt% and 93.04 wt%, respectively. Cai et al. [126] used Fe/Al_2_O_3_ catalysts for catalytic pyrolysis of WPP in a two-stage fixed-bed reactor at 500~800 °C; they found that the purity of CNTs with a diameter of 8~20 nm and high graphitization degree increased with increasing Fe content and the inner diameter varied from a few to dozens of nanometers.

## 3. Challenges and Developments of Co-Pyrolysis for Producing High-Value Carbons

The co-pyrolysis technology of biomass and waste plastics has been studied by many researchers. At present, mature technology in this field has been used to produce light oil, wax, and alkanes, but the co-pyrolysis technology whose target product is high-value carbon materials is still in the laboratory stage. The synthesized high-value carbons can be characterized by scanning electron microscopy, differential scanning calorimetry, Raman spectroscopy, and other instruments. However, the large-scale production of these products is difficult, and it is still necessary to develop new routes to solve the shortcomings [127,128].

Co-pyrolysis of biomass materials and waste plastics is a potentially promising technology for the large-scale synthesis of high-value carbons. The feedstocks come from a variety of sources and can be obtained almost anywhere. For graphitic carbon, the current mature technology is to carbonize biomass materials and waste plastics at a low temperature range and then graphitize at a high temperature. The pyrolysis temperature and graphitization cost can be significantly reduced by using metal catalysts and the quality of graphite products can be improved. However, this technology still has limitations in producing high-quality graphitic carbon. For graphene, FJH can produce carbon materials in a shorter time with high efficiency, but it is still in the research stage. There is little research on the co-pyrolysis of biomass and waste plastics, so the optimal parameters of FJH should be addressed in the future. For biochar and ACs, the yield and performance of carbon materials need to be further improved. For CNTs, further work will focus on the process of gas reforming and the regulation of CNT morphology using pyrolysis catalysts and carbon deposition catalysts.

Based on the understanding of the co-pyrolysis technology and mechanism of waste plastics and waste biomass materials, ideas are provided for the study of co-pyrolysis of waste plastics and biomass materials. For example, polyolefin materials are more suitable for the pyrolysis synthesis of CNTs and other carbon nanoparticles, while PET is more suitable for the synthesis of porous carbons, which opens an important consideration for the research of co-pyrolysis. Polyolefin materials are more likely to generate pyrolysis gas at high temperature and can be used as a carbon source for carbon deposition to synthesize CNTs. The gas generated after PET pyrolysis escapes and many pores are formed in the carbon products, and then the oxygen-containing functional groups on them can improve the adsorption performance through electrostatic interaction and hydrogen bonding.

The carbon materials obtained by co-pyrolysis of biomass materials and waste plastics, such as graphene and CNTs, can be directly used in the production of composite materials. In addition, graphite materials and ACs with porous structure and high surface area can also be applied in wastewater treatment and air purification [129]. Due to their excellent conductivity, these products have good prospects for use as electrode materials for stored energy devices. In the future, we should not only focus on improving the yield of high-value carbon materials in co-pyrolysis, but also realize the efficient utilization and commercialization of carbon products.

## 4. Conclusions

Currently, traditional plastics such as polyolefin and polyester continue to hold a significant position in packaging products. While new packaging materials are constantly being developed, it will take time for these materials to replace traditional ones. This is a hot topic due to the heavy consumption of these plastics and the need for high-value conversion technology for plastic packaging wastes in the future. The review summarizes co-pyrolysis routes of waste plastics and biomass materials with typical polyolefin materials and polyester to produce high-value carbons. It is important to efficiently utilize plastic waste and convert it into high-value products. Carbon materials, including biochar, graphene, graphitic carbon, activated carbon, and carbon nanotubes with high quality and excellent performances, can be obtained by co-pyrolysis of different types of biomass materials and plastics. During this process, the dosage ratio of raw materials, pyrolysis temperature, equipment type, and catalyst species will have varying impacts on the pyrolysis products. Therefore, the optimization of process parameters is the future development direction. However, achieving mass production of high-value carbons through the co-pyrolysis of biomass materials and waste plastics requires continuous development to improve the yield and performance of carbon materials.

## Figures and Tables

**Figure 1 polymers-16-01066-f001:**
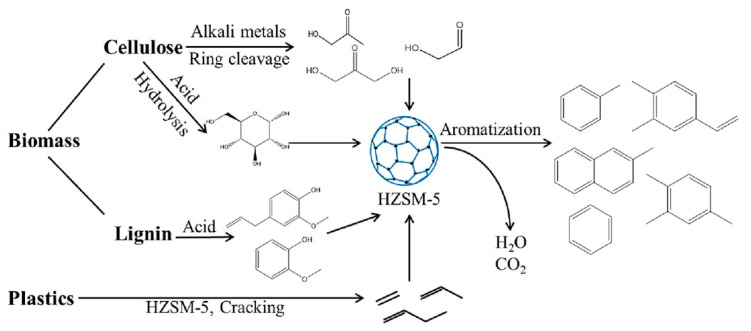
The proposed co-pyrolysis mechanism of biomass and plastics. Reproduced with permission from ref. [39]. Copyright 2023 Elsevier.

**Figure 2 polymers-16-01066-f002:**
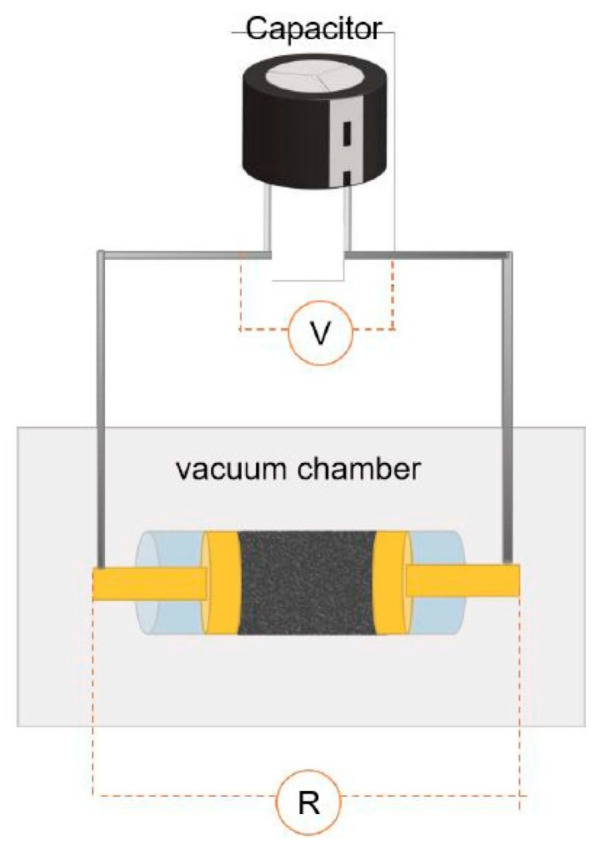
Simplified scheme of the flash Joule heating (FJH) system. Reproduced with permission from ref. [14]. Copyright 2022 Elsevier.

**Figure 3 polymers-16-01066-f003:**
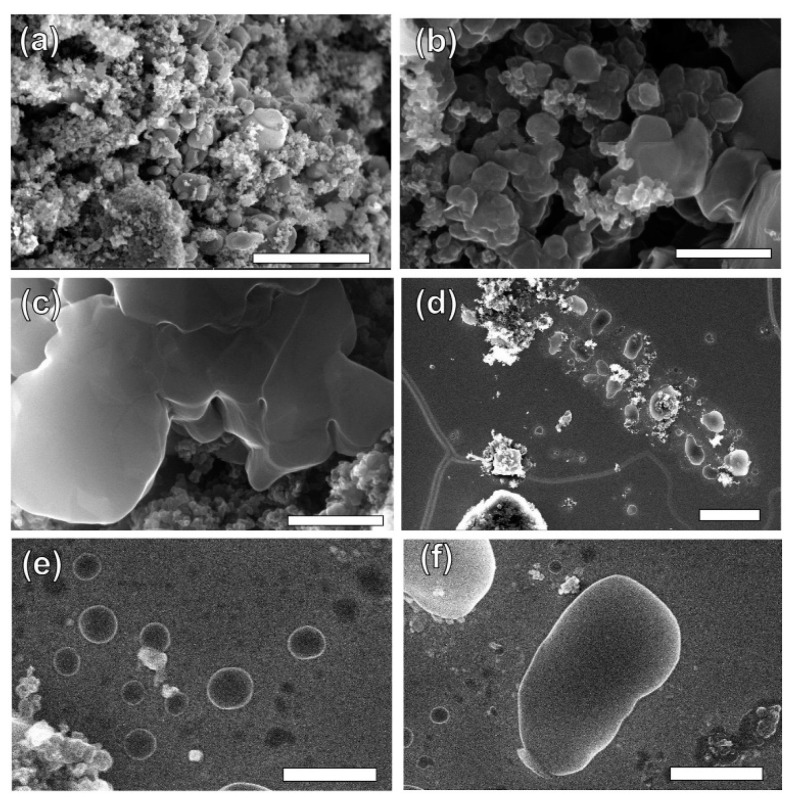
SEM images of (**a**) wrinkled graphene and (**b**,**c**) tFG crystals. Scale bars are 3 μm, 500 nm, and 500 nm, respectively. (**d**) SEM image showing the exfoliation of tFG sheets onto a Au-coated wafer. Scale bar is 5 μm. (**e**) Smaller exfoliated FG flakes of size ~200–300 nm. Scale bar is 500 nm. (**f**) A large tFG platelet. Scale bar is 1 μm. Reproduced with permission from ref. [73]. Copyright 2021 Elsevier.

**Figure 4 polymers-16-01066-f004:**
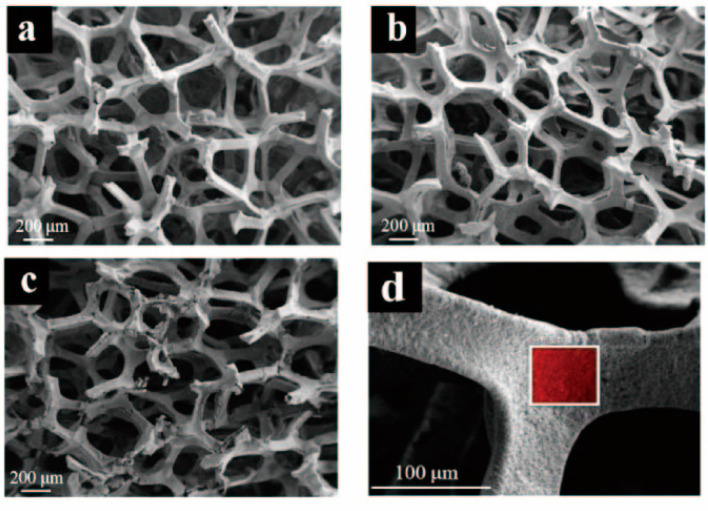
The SEM images of the (**a**) 3DGF_700_, (**b**) 3DGF_800_, and (**c**) 3DGF_900_ after etching the nickel foam. (**d**) FESEM image of 3DGF800. Reproduced with permission from ref. [86]. Copyright 2021 Elsevier.

**Figure 5 polymers-16-01066-f005:**
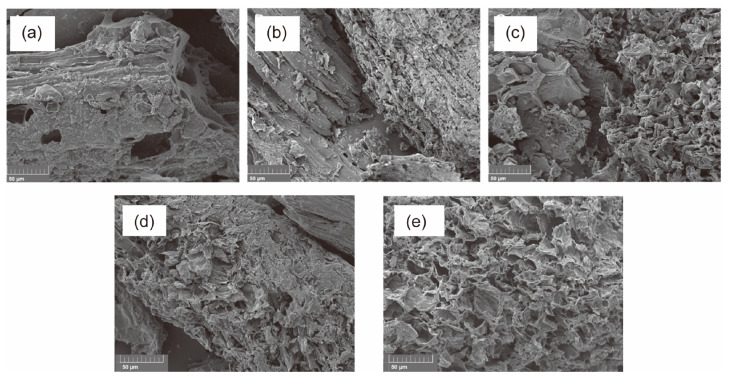
SEM images of select AC samples as a function of the char obtained from the pyrolysis of CS and plastics in various mass ratios. (**a**) AC from CS:PET 1:1, (**b**) AC from CS:PET 4:1, (**c**) AC from CS:PET 9:1, (**d**) AC from CS:PS 4:1, and (**e**) AC from CS:PS 9:1. Reproduced with permission from ref. [96]. Copyright 2023 ACS Omega.

**Figure 6 polymers-16-01066-f006:**
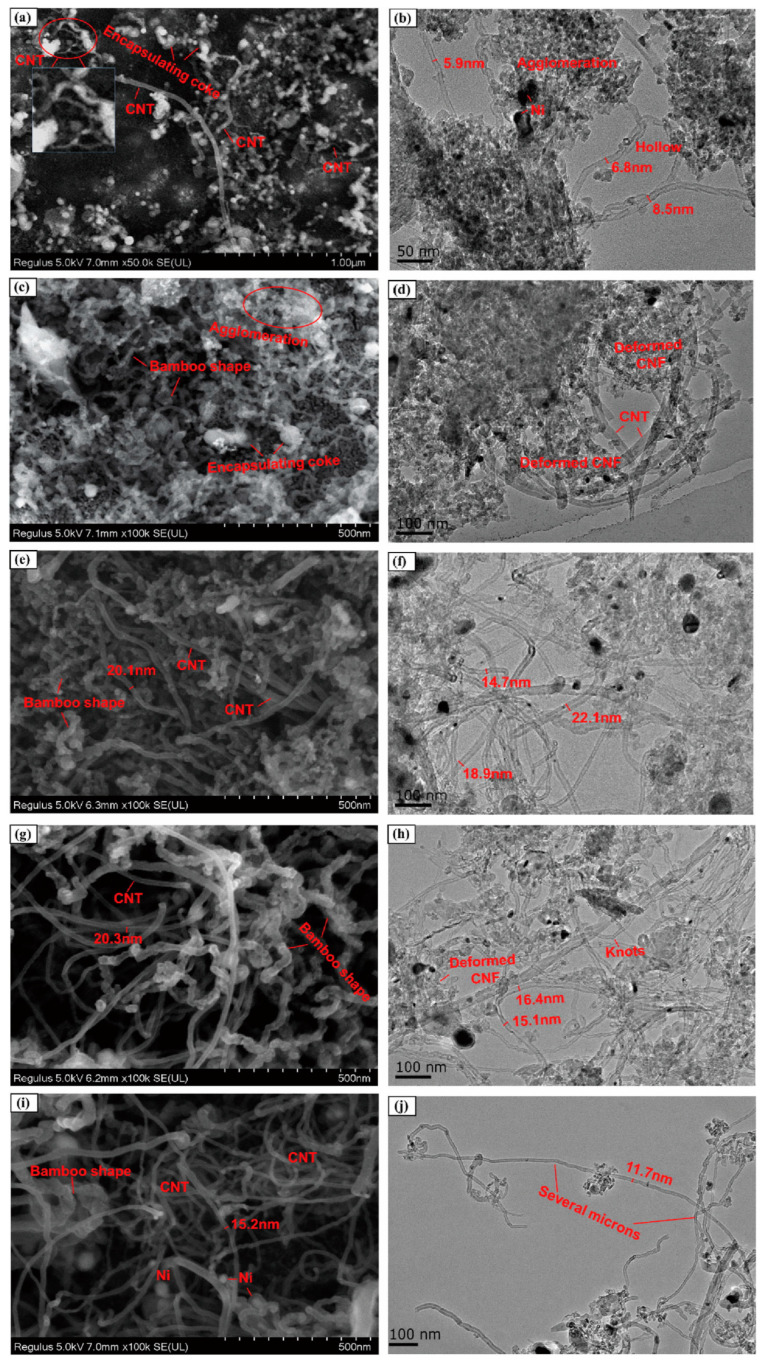
SEM and TEM images of the deposited carbon in different feedstock mixtures on the used Ni/Al_2_O_3_ catalyst: (**a**,**b**) pure RH; (**c**,**d**) 25% PE; (**e**,**f**) 50% PE; (**g**,**h**) 75% PE; (**i**,**j**) pure PE. Reproduced with permission from ref. [118]. Copyright 2021 Elsevier.

## Data Availability

Not applicable.

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
