# Peer review of "Recent Progresses in Pyrolysis of Plastic Packaging Wastes and Biomass Materials for Conversion of High-Value Carbons: A Review"

_polymers, 2024, doi:10.3390/polym16081066_

Round 1
Reviewer 1 Report
Comments and Suggestions for Authors
This is an interesting subject, and I think a valid topic for review. However:
- The motivation for pyrolysis to carbons (vs syngas, simple combustion, or chemical or mechanical recycling) should be articulated more clearly throughout, as should the advantages of using a mixed-waste stream over biomass alone for these applications.
- Discussion of graphene production should be more critical. It is not clear as written whether yields of graphene really refer to genuine isolated/isolable monolayer graphene, or simply carbon deemed suitable for subsequent exfoliation. This makes it hard for the reader to evaluate what exactly is being produced in these studies.
- Figure 4 is not well explained in the text.
- Key parameters for activated carbon porosity (surface area, pore size distribution) are not really discussed specifically, so the reader cannot really evaluate or compare the different approaches discussed. How do these compare to conventional activated charcoals? What are the sustainability advantages to these approaches? More detail needed.
- The section on CNT production needs fleshing out - the applications and description of what is going on in some of the reviewed approaches is a bit opaque and would benefit from more discussion.
I would advise a re-draft to include:
More detail of the motivation/rationale/context behind this approach overall.
More critical discussion of the key parameters of the materials produced, their applications, and the "green-ness" of the method. The reader needs to understand why this technology is a good fit for the application, and what exactly has been produced in terms of both the key quality parameters for the product in question and how it compares to alternatives.
Comments on the Quality of English Language
Whilst for the most part I understood what was meant, the English has numerous minor errors which could be improved by thorough proof-reading to significantly improve readability.
Reviewer 2 Report
Comments and Suggestions for Authors
The article nicely summerize the issue of waste management via pyrolisis. I would like to suggest that the authors may shed some light on the cost of pyrolysis and the likely air quality emissions that it will result in. Often the pyrolysis is not done due to these two important factors.
Comments on the Quality of English LanguageLanguage is fine. The article reads well.
Reviewer 3 Report
Comments and Suggestions for Authors
The authors of the manuscript presented an analysis of a wide range of papers related to the co-pyrolysis of biomass and plastics. The work focuses on the properties of the resulting carbon residue and its variations. Below is a list of comments that may help improve this work.
Row 55. What methods or groups of methods exist besides chemical ones, which include pyrolysis. Some clarification required.
Rows 73-75. The above sources describe improving soil nutrients, but do not describe other applications of biochar. Please add relevant references.
Rows 76-78. Please add reference.
Section 1. Add references and explanations regarding existing reviews on this and related topics. For example, there is a large work doi:10.1016/j.jclepro.2020.121762 devoted to biochar obtained through the process of co-pyrolysis and its applications.
Rows 94-95. “Co-pyrolysis is a process where multiple feedstocks undergo thermal decomposition without air or oxygen”. Perhaps the more common formulation “absence of oxygen” should be used.
Figure 1. Rows 110-112. The appearance of a catalyst is not entirely clear; if the work is devoted to co-pyrolysis in general, is the presence of a catalyst necessary for the implementation of these processes?
Rows 118-120. Please replace reference [31] with an English one.
Rows 122-123. Commercial charcoal and biochar produced by pyrolysis or co-pyrolysis are derived from the same groups of lignocellulosic biomass. On the contrary, agricultural waste often has a low calorific value. Some clarification needed.
Row 133. Use the same expression - replace “carbon” with “biochar”.
Rows 153-154. Reference [41] presents research that is not related to co-pyrolysis, so it is not entirely relevant.
Row 234. The first mention of the acronym HER, whether it means hydrogen evolution reaction, requires clarification.
What is the fundamental difference from the authors' point of view between graphitic carbon and biochar (Sections 2.3.1 and 2.1, respectively)? How are these products different? For example, in [68] (rows 259-263), the co-pyrolysis process was carried out at a temperature of 420 °C and chars yields of 97% were caused by incomplete decomposition. How does this carbon residue relate to carbon?
Section 2.4. Row 318. How do the authors differentiate between activated carbon and biochar? Is this distinction directly related to the use of the material or are there any values for the properties presented in this section?
Round 2
Reviewer 1 Report
Comments and Suggestions for Authors
This is significantly improved with respect to the original draft. The authors have addressed each comment well.
Comments on the Quality of English LanguageAdditional proof-reading by a native or very confident english-speaking chemist would make this much more readable.
Author Response
Response: Thank you for your comment. We did most of the proof-reading on the manuscript.Please see the attachment.
